# A Synaptical Story of Persistent Activity with Graded Lifetime in a Neural System

**Yuanyuan Mi,     Luozheng Li**
State Key Laboratory of Cognitive Neuroscience & Learning,
Beijing Normal University, Beijing 100875, China
miyuanyuan0102@163.com,   liluozheng@mail.bnu.edu.cn

**Dahui Wang**
State Key Laboratory of Cognitive Neuroscience & Learning,
School of System Science, Beijing Normal University,Beijing 100875, China
wangdh@bnu.edu.cn

**Si Wu**
State Key Laboratory of Cognitive Neuroscience & Learning,
IDG/McGovern Institute for Brain Research,
Beijing Normal University ,Beijing 100875, China
wusi@bnu.edu.cn

## Abstract

Persistent activity refers to the phenomenon that cortical neurons keep firing even after the stimulus triggering the initial neuronal responses is moved. Persistent activity is widely believed to be the substrate for a neural system retaining a memory trace of the stimulus information. In a conventional view, persistent activity is regarded as an attractor of the network dynamics, but it faces a challenge of how to be closed properly. Here, in contrast to the view of attractor, we consider that the stimulus information is encoded in a marginally unstable state of the network which decays very slowly and exhibits persistent firing for a prolonged duration. We propose a simple yet effective mechanism to achieve this goal, which utilizes the property of short-term plasticity (STP) of neuronal synapses. STP has two forms, short-term depression (STD) and short-term facilitation (STF), which have opposite effects on retaining neuronal responses. We find that by properly combining STF and STD, a neural system can hold persistent activity of graded lifetime, and that persistent activity fades away naturally without relying on an external drive. The implications of these results on neural information representation are discussed.

## 1   Introduction

Stimulus information is encoded in neuronal responses. Persistent activity refers to the phenomenon that cortical neurons keep firing even after the stimulus triggering the initial neural responses is removed [1, 2, 3]. It has been widely suggested that persistent activity is the substrate for a neural system to retain a memory trace of the stimulus information [4]. For instance, in the classical delayed-response task where an animal needs to memorize the stimulus location for a given period of time before taking an action, it was found that neurons in the prefrontal cortex retained high-frequency firing during this waiting period, indicating that persistent activity may serve as the

neural substrate of working memory [2]. Understanding the mechanism of how persistent activity is generated in neural systems has been at the core of theoretical neuroscience for decades [5, 6, 7].

In a conventional view, persistent activity is regarded as an emergent property of network dynamics: neurons in a network are reciprocally connected with each other via excitatory synapses, which form a positive feedback loop to maintain neural responses in the absence of an external drive; and meanwhile a matched inhibition process suppresses otherwise explosive neural activities. Mathematically, this view is expressed as the dynamics of an attractor network, in which persistent activity corresponds to a stationary state (i.e., an attractor) of the network. The notion of attractor dynamics is appealing, which qualitatively describes a number of brain functions, but its detailed implementation in neural systems remains to be carefully evaluated.

A long-standing debate on the feasibility of attractor dynamics is on how to properly close the attractor states in a network: once a neural system is evolved into a self-sustained active state, it will stay there forever until an external force pulls it out. Solutions including applying a strong global inhibitory input to shut-down all neurons simultaneously, or applying a strong global excitatory input to excite all neurons and force them to fall into the refractory period simultaneously, were suggested [9], but none of them appears to be natural or feasible in all conditions. From the computational point of view, it is also unnecessary for a neural system to hold a mathematically perfect attractor state lasting forever. In reality, the brain only needs to hold the stimulus information for a finite amount of time necessary for the task. For instance, in the delayed-response task, the animal only needed to memorize the stimulus location for the waiting period [1].

To address the above issues, here we propose a novel mechanism to retain persistent activity in neural systems, which gives up the concept of prefect attractor, but rather consider that a neural system is in a marginally unstable state which decays very slowly and exhibits persistent firing for a prolonged period. The proposed mechanism utilizes a general feature of neuronal interaction, i.e., the short-term plasticity (STP) of synapses [10, 11]. STP has two forms: short-term depression (STD) and short-term facilitation (STF). The former is due to depletion of neurotransmitters after neural firing, and the latter is due to elevation of calcium level after neural firing which increases the release probability of neurotransmitters. STD and STP have opposite effects on retaining prolonged neuronal responses: the former weakens neuronal interaction and hence tends to suppress neuronal activities; whereas, the latter strengthens neuronal interaction and tends to enhance neuronal activities. Interestingly, we find that the interplay between the two processes endows a neural system with the capacity of holding persistent activity with desirable properties, including: 1) the lifetime of persistent activity can be arbitrarily long depending on the parameters; and 2) persistent activity fades away naturally in a network without relying on an external force. The implications of these results on neural information representation are discussed.

## 2 The Model

Without loss of generality, we consider a homogeneous network in which neurons are randomly and sparsely connected with each other with a small probability $p$. The dynamics of a single neuron is described by an integrate-and-fire process, which is given by

$$\tau \frac{dv_i}{dt} = -(v_i - V_L) + R_m h_i, \quad \text{for } i = 1 \dots N, \tag{1}$$

where $v_i$ is the membrane potential of the $i$th neuron and $\tau$ the membrane time constant. $V_L$ is the resting potential. $h_i$ is the synaptic current and $R_m$ the membrane resistance. A neuron fires when its potential exceeds the threshold, i.e., $v_i > V_{th}$, and after that $v_i$ is reset to be $V_L$. $N$ the number of neurons.

The dynamics of the synaptic current is given by

$$\tau_s \frac{dh_i}{dt} = -h_i + \frac{1}{Np} \sum_j J_{ij} u_j^+ x_j^- \delta(t - t_j^{sp}) + I^{ext} \delta(t - t_i^{ext}), \tag{2}$$

where $\tau_s$ is the synaptic time constant, which is about $2 \sim 5$ms. $J_{ij}$ is the absolute synaptic efficacy from neurons $j$ to $i$. $J_{ij} = J_0$ if there is a connection from the neurons $j$ to $i$, and $J_{ij} = 0$ otherwise. $t_j^{sp}$ denotes the spiking moment of the neuron $j$. All neurons in the network receive an external input

in the form of Poisson spike train. $I^{ext}$ represents the external input strength and $t_i^{ext}$ the moment of the Poisson spike train the neuron $i$ receives.

The variables $u_j$ and $x_j$ measure, respectively, the STF and STD effects on the synapses of the $j$th neuron, whose dynamics are given by [12, 13]

$$\tau_f \frac{du_j}{dt} = -u_j + \tau_f U(1 - u_j^-)\delta(t - t_j^{sp}), \tag{3}$$

$$\tau_d \frac{dx_j}{dt} = 1 - x_j - \tau_d u_j^+ x_j^- \delta(t - t_j^{sp}), \tag{4}$$

where $u_j$ is the release probability of neurotransmitters, with $u_j^+$ and $u_j^-$ denoting, respectively, the values of $u_j$ just after and just before the arrival of a spike. $\tau_f$ is the time constant of STF. $U$ controls the increment of $u_j$ produced by a spike. Upon the arrival of a spike, $u_j^+ = u_j^- + U(1 - u_j^-)$. $x_j$ represents the fraction of available neurotransmitters, with $x_j^+$ and $x_j^-$ denoting, respectively, the values of $x_j$ just after and just before the arrival of a spike. $\tau_d$ is the recover time of neurotransmitters. Upon the arrival of a spike, $x_j^+ = x_j^- - u_j^+ x_j^-$. The time constants $\tau_f$ and $\tau_d$ are typically in the time order of hundreds to thousands of milliseconds, much larger than $\tau$ and $\tau_s$, that is, STP is a slow process compared to neural firing.

## 2.1  Mean-field approximation

As to be confirmed by simulation, neuronal firings in the state of persistent activity are irregular and largely independent to each other. Therefore, we can assume that the responses of individual neurons are statistically equivalent in the state of persistent activity. Under this mean-field approximation, the dynamics of a single neuron, and so does the mean activity of the network, can be written as [7]

$$\tau_s \frac{dh}{dt} = -h + J_0 uxR + I, \tag{5}$$

$$\tau_f \frac{du}{dt} = -u + \tau_f U(1 - u)R, \tag{6}$$

$$\tau_d \frac{dx}{dt} = 1 - x - \tau_d uxR, \tag{7}$$

where the state variables are the same for all neurons. $R$ is the firing rate of a neuron, which is also the mean activity of the neuron ensemble. $I = I^{ext}\lambda$ denotes the external input with $\lambda$ the rate of the Poisson spike train. The exact relationship between the firing rate $R$ and the synaptic input $h$ is difficult to obtain. Here, we assume it to be of the form,

$$R = \max(\beta h, 0), \tag{8}$$

with $\beta$ a positive constant.

## 3  The Mechanism

By using the mean-field model, we first elucidate the working mechanism underlying the generation of persistent activity of finite lifetime. Later we carry out simulation to confirm the theoretical analysis.

## 3.1  How to generate persistent activity of finite lifetime

For the illustration purpose, we only study the dynamics of the firing rate $R$ and assume that the variables $u$ and $x$ reach to their steady values instantly. This approximation is in general inaccurate, since $u$ and $x$ are slow variables compared to $R$. Nevertheless, it gives us insight into understanding the network dynamics.

By setting $du/dt = 0$ and $dx/dt = 0$ in Eqs.(6,7) and substituting them into Eqs.(5,8), we get that, for $I = 0$ and $R \geq 0$,

$$\tau_s \frac{dR}{dt} = -R + \frac{J_0 \beta \tau_f U R^2}{1 + \tau_f U R + \tau_d \tau_f U R^2} \equiv F(R). \tag{9}$$

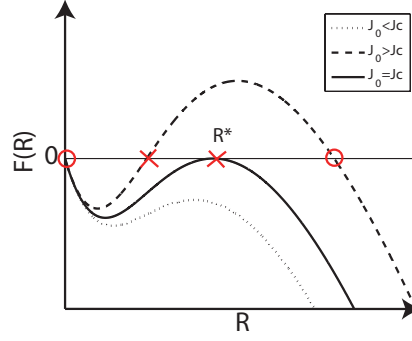

Figure 1: The steady states of the network, i.e., the solutions of Eq.(9), have three forms depending on the parameter values. The three lines correspond to the different neuronal connection strenghths, which are $J_0 = 4, 4.38, 5$, respectively. The other parameters are: $\tau_s = 5ms, \tau_d = 100ms, \tau_f = 700ms, \beta = 1, U = 0.05$ and $J_c = 4.38$.

Define a critical connection strength $J_c \equiv \left(1 + 2\sqrt{\tau_d/(\tau_f U)}\right)/\beta$, which is the point the network dynamics experiences saddle-node bifurcation (see Figure 1). Depending on the parameters, the steady states of the network have three forms

- When $J_0 < J_c$, $F(R) = 0$ has only one solution at $R = 0$, i.e., the network is only stable at the silent state;

- When $J_0 > J_c$, $F(R) = 0$ has three solutions, and the network can be stable at the silent state and an active state;

- When $J_0 = J_c$, $F(R) = 0$ has two solutions, one is the stable silent state, and the other is a neutral stable state, referred to as $R^*$.

The interesting behavior occurs at $J_0 = J_c^-$, i.e., $J_0$ is slightly smaller than the critical connection strength $J_c$. In this case, the network is only stable at the silent state. However, since near to the state $R^*$, $F(R)$ is very close to zero (and so does $|dR/dt|$), the decay of the network activity is very slow in this region (Figure 2A). Suppose that the network is initially at a state $R > R^*$, under the network dynamics, the system will take a considerable amount of time to pass through the state $R^*$ before reaching to silence. This is manifested by that the decay of the network activity exhibits a long plateau around $R^*$ before dropping to silence rapidly (Figure 2B). Thus, persistent activity of finite lifetime is achieved.

The lifetime of persistent activity, which is dominated by the time of the network state passing through the point $R^*$, is calculated to be (see Appendix A),

$$T \sim \frac{2\tau_s}{\sqrt{F(R^*)F''(R^*)}}, \tag{10}$$

where $F''(R^*) = d^2F(R)/d^2R|_{R^*}$. By varying the STP effects, such as $\tau_d$ and $\tau_f$, the value of $F(R^*)F''(R^*)$ is changed, and the lifetime of persistent activity can be adjusted.

## 3.2 Persistent activity of graded lifetime

We formally analyze the condition for the network holding persistent activity of finite lifetime. Inspired by the result in the proceeding section, we focus on the parameter regime of $J_0 = J_c$, i.e., the situation when the network has the stable silent state and a neutral stable active state.

Denote $(R^*, u^*, x^*)$ to be the neutral stable state of the network at $J_0 = J_c$. Linearizing the network dynamics at this point, we obtain

$$\frac{d}{dt}\begin{pmatrix} R - R^* \\ u - u* \\ x - x* \end{pmatrix} \simeq \mathbf{A} \begin{pmatrix} R - R^* \\ u - u* \\ x - x^* \end{pmatrix}, \tag{11}$$

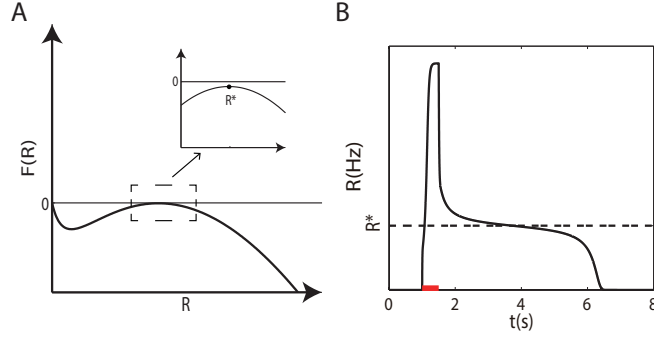

Figure 2: Persistent activity of finite lifetime. Obtained by solving Eqs.(5-8). (A) When $J_0 = J_c^-$, the function $F(R)$, and so does $dR/dt$, is very close to zero at the state $R^*$. Around this point, the network activity decays very slowly. The inset shows the fine structure in the vicinity of $R^*$. (B) An external input (indicated by the red bar) triggers the network response. After removing the external input, the network activity first decays quickly, and then experiences a long plateau before dropping to silence rapidly. The parameters are: $\tau_s = 5ms, \tau_d = 10ms, \tau_f = 800ms, \beta = 1, U = 0.5, I = 10, J_c = 1.316$ and $J_0 = 1.315$.

where $\mathbf{A}$ is the Jacobian matrix (see Appendix B).

It turns out that the matrix $\mathbf{A}$ always has one eigenvector with vanishing eigenvalue, a property due to that $(R^*, u^*, x^*)$ is the neutral stable state of the network dynamics. As demonstrated in Sec.3.1, by choosing $J_0 = J_c^-$, we expect that the network state will decay very slowly along the eigenvector of vanishing eigenvalue, which we call the decay-direction. To ensure this always happens, it requires that the real parts of the other two eigenvalues of $\mathbf{A}$ are negative, so that any perturbation of the network state away from the decay-direction will be pulled back; otherwise, the network state may approach to silence rapidly via other routes avoiding the state $(R^*, u^*, x^*)$. This idea is illustrated in Fig.3.

The condition for the real parts of the other two eigenvalues of $\mathbf{A}$ being smaller than zero is calculated to be (see Appendix B):

$$\frac{2}{\tau_f \tau_d} + \frac{1}{\tau_d}\sqrt{\frac{U}{\tau_f \tau_d}} + \frac{1}{\tau_d \tau_s}\frac{1}{1+\sqrt{\frac{\tau_f U}{\tau_d}}} - \frac{1}{\tau_f \tau_s} > 0. \tag{12}$$

This inequality together with $J_0 = J_c^-$ form the condition for the network holding persistent activity of finite lifetime.

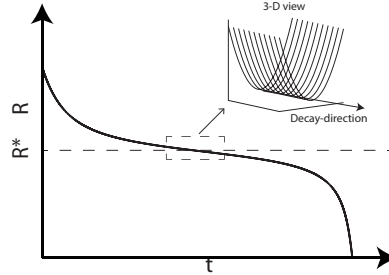

Figure 3: Illustration of the slow-decaying process of the network activity. The network dynamics experiences a long plateau before dropping to silence quickly. The inset presents a 3-D view of the local dynamics in the plateau region, where the network state is attracted to the decay-direction to ensure slow-decaying.

By solving the network dynamics Eqs.(5-8), we calculate how the lifetime of persistent activity changes with the STP effect. Fig.4A presents the results of fixing $U$ and $J_0$ and varying $\tau_d$ and

$\tau_f$, We see that below the critical line $J_0 = J_c$, which is the region for $J_0 > J_c$, the network has prefect attractor states never decaying; and above the critical line, the network has only the stable silent state. Close to the critical line, the network activity decays slowly and displays persistent activity of finite lifetime. Fig.4B shows a case that when the STF strength ($\tau_f$) is fixed, the lifetime of persistent activity decreases with the STD strength ($\tau_d$). This is understandable, since STD tends to suppress neuronal responses. Fig.4C shows a case that when $\tau_d$ is fixed, the lifetime of persistent activity increases with $\tau_f$, due to that STF enhances neuronal responses. These results demonstrate that by regulating the effects of STF and STD, the lifetime of persistent activity can be adjusted.

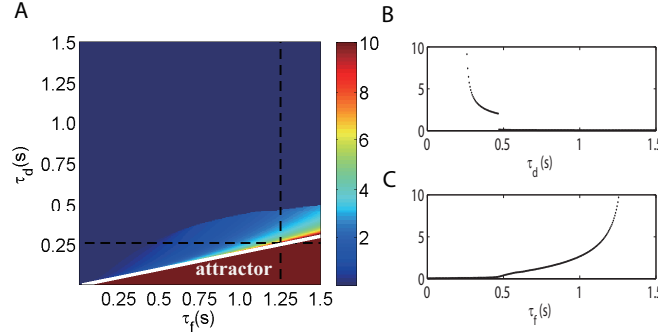

Figure 4: (A). The lifetimes of the network states with respect to $\tau_f$ and $\tau_d$ when $U$ and $J_0$ are fixed. We use an external input to trigger a strong response of the network and then remove the input. The lifetime of a network state is measured from the offset of the external input to the moment when the network returns to silence. The white line corresponds to the condition of $J_0 = J_c$, below which the network has attractors lasting forever; and above which, the lifetime of a network state gradually decreases (coded by colour). (B) When $\tau_f = 1250ms$ is fixed, the lifetime of persistent activity decreases with $\tau_d$ (the vertical dashed line in A). (C) When $\tau_d = 260ms$ is fixed, the lifetime of persistent activity increases with $\tau_f$ (the horizontal dashed line in A). The other parameters are: $\tau_s = 5ms$, $\beta = 1$, $U = 0.05$ and $J_0 = 5$.

## 4  Simulation Results

We carry out simulation with the spiking neuron network model given by Eqs.(1-4) to further confirm the above theoretical analysis. A homogenous network with $N = 1000$ neurons is used, and in the network neurons are randomly and sparsely connected with each other with a probability $p = 0.1$. At the state of persistent activity, neurons fire irregularly (the mean value of Coefficient of Variation is 1.29)and largely independent to each other(the mean correlation of all spike train pairs is 0.30) with each other (Fig.5A). Fig.5 present the examples of the network holding persistent activity with varied lifetimes, through different combinations of STF and STD satisfying the condition Eq.(12).

## 5  Conclusions

In the present study, we have proposed a simple yet effective mechanism to generate persistent activity of graded lifetime in a neural system. The proposed mechanism utilizes the property of STP, a general feature of neuronal synapses, and that STF and STD have opposite effects on retaining neuronal responses. We find that with properly combined STF and STD, a neural system can be in a marginally unstable state which decays very slowly and exhibits persistent firing for a finite lifetime. This persistent activity fades away naturally without relying on an external force, and hence avoids the difficulty of closing an active state faced by the conventional attractor networks.

STP has been widely observed in the cortex and displays large diversity in different regions [14, 15, 16]. Compared to static ones, dynamical synapses with STP greatly enriches the response patterns and dynamical behaviors of neural networks, which endows neural systems with information processing capacities which are otherwise difficult to implement using purely static synapses. The research on the computational roles of STP is receiving increasing attention in the field [12]. In

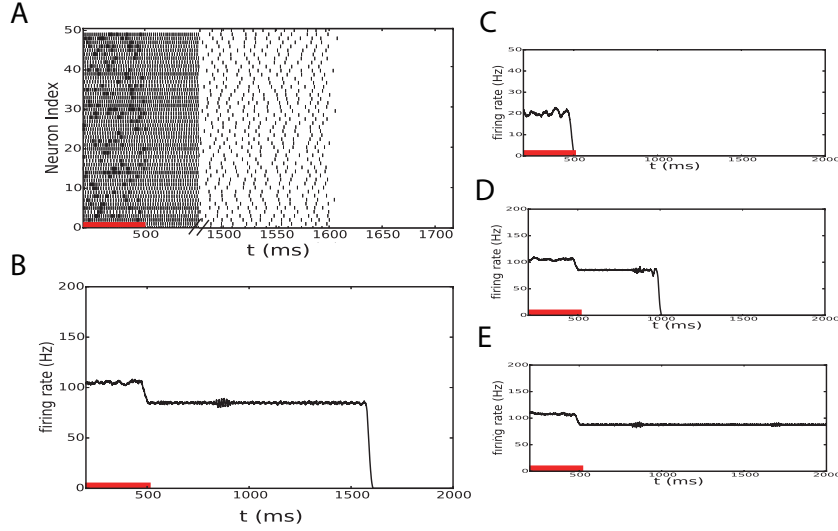

Figure 5: The simulation results of the spiking neural network. (A) A raster plot of the responses of 50 example neurons randomly chosen from the network. The external input is applied for the first 0.5 second. The persistent activity lasts about $1100ms$. The parameters are: $\tau_f = 800ms, \tau_d = 500ms, U = 0.5, J_0 = 28.6$. (B) The firing rate of the network for the case (A). (C) An example of persistent activity of negligible lifetime. The parameters are:$\tau_f = 800ms, \tau_d = 1800ms, U = 0.5, J_0 = 28.6$. (D) An example of persistent activity of around 400ms lifetime. The parameters are:$\tau_f = 600ms, \tau_d = 500ms, U = 0.5, J_0 = 28.6$. (E) An example of the network holding an attractor lasting forever. The parameters are: $\tau_f = 800ms, \tau_d = 490ms, U = 0.5, J_0 = 28.6$.

terms of information presentation, a number of appealing functions contributed by STP were proposed. For instances, Mongillo et al. proposed an economical way of using the facilitated synapses due to STF to realize working memory in the prefrontal cortex without recruiting neural firing [8]; Pfister et al. suggested that STP enables a neuron to estimate the membrane potential information of the pre-synaptic neuron based on the spike train it receives [17]. Torres et al. found that STD induces instability of attractor states in a network, which could be useful for memory searching [18]; Fung et al. found that STD enables a continuous attractor network to have a slow-decaying state in the time order of STD, which could serve for passive sensory memory [19]. Here, our study reveals that through combining STF and STD properly, a neural system can hold stimulus information for an arbitrary time, serving for different computational purposes. In particular, STF tends to increase the lifetime of persistent activity; whereas, STD tends to decrease the lifetime of persistent activity. This property may justify the diverse distribution of STF and STD in different cortical regions. For instances, in the prefrontal cortex where the stimulus information often needs to be held for a long time in order to realize higher cognitive functions, such as working memory, STF is found to be dominating; whereas, in the sensory cortex where the stimulus information will be forwarded to higher cortical regions shortly, STD is found to be dominating. Furthermore, our findings suggest that a neural system may actively regulate the combination of STF and STD, e.g., by applying appropriate neural modulators [10], so that it can hold the stimulus information for a flexible amount of time depending on the actual computational requirement. Further experimental and theoretical studies are needed to clarify these interesting issues.

# 6   Acknowledgments

This work is supported by grants from National Key Basic Research Program of China (NO.2014CB846101), and National Foundation of Natural Science of China (No.11305112, Y.Y.M.; No.31261160495, S.W.; No.31271169,D.H.W.), and the Fundamental Research Funds for the central Universities (No.31221003, S.W.), and SRFDP (No.20130003110022, S.W), and Natural Science Foundation of Jiangsu Province BK20130282.

## Appendix A: The lifetime of persistent activity

Consider the network dynamics Eq.(9). When $J_0 = J_c$, the network has a stable silent state ($R = 0$) and an unstable active state, referred to as $R^*$ (Fig.1). We consider that $J_0 = J_c^-$. In this case, $F(R^*)$ is slightly smaller than zero (Fig.2A). Starting from a state $R > R^*$, the network will take a considerable amount of time to cross the point $R^*$, since $dR/dt$ is very small in this region, and the network exhibits persistent activity for a considerable amount of time. We estimate the time consuming for the network crossing the point $R^*$.

According to Eq.(9), we have

$$
\begin{aligned}
\int_0^T dt &= \int_{R_-^*}^{R_+^*} \frac{\tau_s}{F(R)} dR \approx \int_{R_-^*}^{R_+^*} \frac{\tau_s dR}{F(R^*) + (R - R^*)^2 F''(R^*)/2}, \\
&= \frac{2\tau_s}{\sqrt{F(R^*)F''(R^*)}} \left[ \operatorname{arctg} \frac{R_+^* - R^*}{\sqrt{F(R^*)/F''(R^*)}} - \operatorname{arctg} \frac{R_-^* - R^*}{\sqrt{F(R^*)/F''(R^*)}} \right], \\
&= \frac{2\tau_s}{\sqrt{F(R^*)F''(R^*)}} G(R^*),
\end{aligned}
\tag{13}
$$

where $R_+^*$ and $R_-^*$ denote, respectively, the points slightly larger or smaller than $R^*$, $F'(R^*) = dF(R)/dR|_{R^*}$, and $F''(R^*) = dF'(R)/dR|_{R^*}$. To get the above result, we used the second-order Taylor expansion of $F(R)$ at $R^*$, and the condition $F'(R^*) = 0$.

In the limit of $F(R^*) \to 0$, the value of $G(R^*)$ is bounded. Thus, the lifetime of persistent activity is in the order of

$$
T \sim \frac{2\tau_s}{\sqrt{F(R^*)F''(R^*)}}.
\tag{14}
$$

## Appendix B: The condition for the network holding persistent activity of finite lifetime

Denote $(R^*, u^*, x^*)$ to be the neutral stable state of the network when $J_0 = J_c$, which is calculated to be (by solving Eqs.(5-8)),

$$
R^* = \sqrt{1/\tau_f \tau_d U}, \quad u^* = \frac{\tau_f U R^*}{1 + \tau_f U R^*}, \quad x^* = \frac{1 + \tau_f U R^*}{1 + \tau_f U R^* + \tau_f \tau_d U R^{*2}}.
\tag{15}
$$

Linearizing the network dynamics at this point, we obtain Eq.(12), in which the Jacobian matrix $\mathbf{A}$ is given by

$$
\mathbf{A} = \begin{pmatrix} (J_0 u^* x^* - 1)/\tau_s, & J_0 x^* R^*/\tau_s, & J_0 u^* R^*/\tau_s \\ U(1 - u^*), & -1/\tau_f - U R^*, & 0 \\ -u^* x^*, & -x^* R^*, & -1/\tau_d - u^* R^* \end{pmatrix}.
\tag{16}
$$

The eigenvalues of the Jacobian matrix satisfy the equality $|\mathbf{A} - \lambda \mathbf{I}| = 0$. Utilizing Eqs.(15), this equality becomes

$$
\lambda(\lambda^2 + b\lambda + c\lambda) = 0,
\tag{17}
$$

where the coefficients $b$ and $c$ are given by

$$
b = \frac{1}{\tau_d} + \frac{1}{\tau_f} + u^* R^* + U R^*,
\tag{18}
$$

$$
c = \frac{2}{\tau_f \tau_d} + \frac{1}{\tau_d} \sqrt{\frac{U}{\tau_f \tau_d}} + \frac{1}{\tau_d \tau_s} \frac{1}{1 + \sqrt{\frac{\tau_f U}{\tau_d}}} - \frac{1}{\tau_f \tau_s}.
\tag{19}
$$

From Eq.(17), we see that the matrix $\mathbf{A}$ has three eigenvalues. One eigenvalue, referred to as $\lambda_1$, is always zero. The other two eigenvalues satisfy that $\lambda_2 + \lambda_3 = -b$ and $\lambda_2 \lambda_3 = c$. Since $b > 0$, the condition for the real parts of $\lambda_2$ and $\lambda_3$ being negative is $c > 0$.

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
