[Reviews · NeurIPS 2014]

Submitted by Assigned_Reviewer_11

Summary: The authors present a model of persistent activity in neural networks as typically observed in working memory tasks and modeled as attractor dynamics. The authors note the shortcomings of existing models, namely the reliance on implausible global excitatory or inhibitory signals to reset the network dynamics after settling into an attractor state and the superfluousness of such a stable attractor, since in working memory tasks, the dynamics need only persistent as long as the task requires, not indefinitely. In the proposed model, these issues are confronted by incorporating short term synaptic facilitation and depression into a network model. Using a mean-field approach, the authors identify stable fixed points of the rate dynamics and how these fixed-points change as a function of network connectivity and timescale parameters. Using this formalism, the authors identify a family of parameters for which the network exhibits only 1 true stable fixed point- corresponding to network quiescence- but also exhibits a second, "slow point", i.e. a point which under different network parameters would correspond to a 2nd unstable fixed point of the rate dynamics, but for the chosen parameters yields network dynamics that evolve extremely slowly towards the true, stable fixed point. The primary effect of this “slow point” is to create a quasi-stable firing rate, similar to the attractor dynamics of existing models, but different in that the dynamics will eventually collapse to the quiescent state. The authors illustrate the existence of this state over a range of timescales of the synaptic plasticity and instantiate the presented framework in a full, spiking neural network.

Review: This is an exceptionally clear-written submission of great interest to the theoretical neuroscience community. It stands out compared to alternative work in attractor dynamics in that it assigns a useful computation role for short-term synaptic plasticity in generating these dynamics and for noting that real biological systems have limited use for “true” fixed-points. The authors eloquently note that neural dynamics need only remain stable for finite, not infinite time, and propose a biologically plausible mechanism to both generate and terminate this quasi-stability. The reduced, mean-field model is clearly presented and illustrates quite nicely the relationship between network parameters and the stability of the slow dynamics. All the figures positively contribute to the submission and create a simple visual representation of the text’s mathematical details. The authors do a great job of explaining the opposing roles facilitation and depression play in generating the slow network dynamics, as most clearly displayed in Figure 4.

All-around a solid submission, however, two small points for improvement. 1. The authors claim that the spiking model exhibits irregular, independent activity (critical to invoke a mean-field assumption) which is difficult to see in Figure 5a, and upon closer inspection, appears to be untrue. As best as I can tell, the spike trains appear extremely regular, and even begin to synchronize a bit (as can be seen in the firing rate, around 2.75 seconds in Figure 5B). The main analytical results seem to hold true as presented here, but rely on this assumption. Clearly displaying the irregularity and independence in the spiking simulations would improve the submission. 2. The main analytical results are derived under the assumption that the synaptic plasticity variables have reached a steady-state, despite the fact that, as the authors note, these variables evolve slowly compared to the rate. A discussion of the conditions under which this assumption is valid would be helpful.
Summary: For a network model of persistent activity that includes both short term synaptic facilitation and depression, the authors analytically compute network connectivity and time constant parameters such that the network dynamics exhibit quasi-stable attractor dynamics over long timescales, but that eventually decay to quiescence. The proposed solution is instantiated in a spiking neural network and is consistent with observed experimental findings.

Submitted by Assigned_Reviewer_12

In the manuscript "A Synaptical Story of Persistent Activity with Graded Lifetime in a Neural System" the authors investigate the effect of synaptic Short Term Plasticity (STP) on marginally unstable states of spiking neuronal networks. A problem of attractor models of memory is that neuronal networks proposed so far do not leave the attractor state on their own. First they calculate stability conditions for a mean-field approximation of the dynamics, derive conditions for slightly unstable dynamics, and show the same effect in networks of leaky integrate-and-fire neurons. The catch is that the interplay of short term depression and short term facilitation allow to tune parameters in a way that an activity state enforced on the network decays in principle arbitrarily slow.

The manuscript is well written and easy to understand, and the presentation of the results is good.

Summary: A well written paper that presents a synaptic mechanism for attractor neuronal networks to leave attractor states after almost arbitrary long times.

Submitted by Assigned_Reviewer_33

In this study, the lifetimes of transient activity states in recurrent networks with short term synaptic plasticity (STP) are studied both, analytically and by simulations. Not surprisingly, fascilitation and depression contribute significantly and by a careful choice of the parameters the system can retain activity for long times. It is not clear how the brain could adapt these parameters in order to achieve a particular lifetime in a network. Reference 14, which layed ground to the mean field approach used in this paper, surprisingly is not cited in the text. Taken together, this work is very well done, the results are solid and moderately contribute to the state of art mainly by its technically clear approach.
Summary: The impact of short term synaptic plasticity on the lifetime of transient activity states in recurrent networks is investigated. While rather straightforeward, the work is well done, the results are well presented and solid.
Author Feedback
Author rebuttal: To Reviewer_ 11:

We acknowledge very much the encouraging comments of the reviewer.

In the simulations, we found that in some parameter regimes for persistent activity, neurons fire irregularly and independently; and in other parameter regimes for persistent activity, neurons display occasional synchronized firings (as presented in Fig.5a). The mean-field approximation works well for the former case. In the latter case, the imaginary parts of the eigenvalues of the Jacob matrix A are non-zero (Eq.18). We should use a proper figure instead of Fig.5a to represents the situation when the mean-field approximation works well.

The assumption that the synaptic plasticity variables have reached a steady-state before the neuronal firing rate does is not accurate in practice. Here, our use of this approximation is only for the convenience of elucidating the network mechanism clearly. Sec.3.2 and Appendix B present the analysis for the full network dynamics without using this assumption.

To Reviewer_12:

We acknowledge the valuable comments of the reviewer.

J_0 used in the sparely connected spiking neuronal network is equivalent to J_0 used in the mean-field approximation (note the normalization term 1/NP in Eq.2).

Yes, all neurons are excited in the simulation.

Here, we consider that a single homogeneous network is to encode a single item; activating a portion of the network, if it is sufficiently large, will excite the whole network and exhibit the same decaying behavior. If we consider that a neural system memorizes several items, then we should include several such homogenous networks and their overlaps reflect the similarities between memorized items.

The paper [20] obtained T~150\tau_s under the condition of \tau_d=60\tau_s,; in other words, with only STD, the network can only hold memory trace in the time order of \tau_d, which is up to several hundreds of milliseconds. The present study proposes a more general mechanism, which shows that with proper combination of STF and STD, the network can hold memory trace up to an arbitral long lifetime.

To Reviewer_33

We acknowledge the valuable comments of the reviewer. Sorry for not citing [14] properly.

We show that by regulating the combination of STF and STD, a neural system can hold the stimulus information for different lifetimes. The brain can manipulate the combination of STF and STD in two different ways: 1) in a single brain region, neural modulator may temporally adjust the effects of STF and STD depending on the computational task [11]; 2) accross brain areas where stimulus information is held for different lifetimes, the combinations of STF and STD are different. This may just the large diversity of STP observed in different cortical regions.